# The Prevalence of Anemia and Its Associated Factors among Older Persons: Findings from the National Health and Morbidity Survey (NHMS) 2015

**DOI:** 10.3390/ijerph19094983

**Published:** 2022-04-20

**Authors:** Ambigga Krishnapillai, Mohd Azahadi Omar, Suthahar Ariaratnam, Smaria Awaluddin, Rajini Sooryanarayana, Ho Bee Kiau, Noorlaili Mohd Tauhid, Sazlina Shariff Ghazali

**Affiliations:** 1Department of Primary Care Medicine, Faculty of Medicine and Defense Health, National Defense University of Malaysia, Kuala Lumpur 57000, Malaysia; 2Institute for Public Health, National Institute of Health, Ministry of Health Malaysia, Setia Alam 40170, Selangor, Malaysia; drazahadi@moh.gov.my (M.A.O.); smaria@moh.gov.my (S.A.); 3Department of Psychiatry, Faculty of Medicine, University Teknologi MARA, Kuala Lumpur 57000, Malaysia; suthaharariaratnam@yahoo.com.au; 4Family Health Development Division, Ministry of Health Malaysia, Putrajaya 62590, Malaysia; drrajini@moh.gov.my; 5Bandar Botanic Health Center, Blok A, Jalan Langat, Bandar Botanik, Ministry of Health Malaysia, Klang 41200, Selangor, Malaysia; drhbk@moh.gov.my; 6Department of Family Medicine, Faculty of Medicine, University Kebangsaan Malaysia, Bangi 43600, Malaysia; lailitauhid@yahoo.com; 7Department of Family Medicine, Faculty of Medicine and Health Sciences, University Putra Malaysia, Serdang 43400, Malaysia; sazlina@upm.edu.my

**Keywords:** anemia, older persons, Malaysia, NHMS, disability, chronic disease

## Abstract

Background: There is limited evidence on the association of anemia with chronic diseases and disabilities among older persons in Malaysia. We assessed the prevalence of anemia and its associated factors among community-dwelling older persons. Methods: This was a cross-sectional study using data from the nationwide National Health and Morbidity Survey 2015 (NHMS 2015) on the health of older adults conducted by the Institute for Public Health, National Institutes of Health, Malaysia. A two-stage stratified random-cluster sampling design was utilized. Data were collected on the sociodemographic profiles, non-communicable disease (NCD) comorbidities (hypertension, diabetes and hypercholesterolemia status) and disabilities among the older persons. Anemia was defined based on the World Health Organization’s standards. A multivariable logistic regression analysis was used to assess the association of anemia with chronic diseases and disabilities. Results: The prevalence of anemia was 35.3% (95% CI: 33.1, 37.4) in the older persons. Chronic disease profiling showed that the prevalence rates of anemia among the older persons with diabetes, hypertension and hypercholesterolemia were 38.6%, 35.3% and 34.1%, respectively. In the multivariable analysis, persons aged 80 years and above (adjusted OR (aOR): 2.64; 95% CI: 2.00, 3.47), 70–79 years (aOR: 1.42; 95% CI: 1.21, 1.66), with diabetes (aOR: 1.30; 95% CI: 1.13, 1.51) and with disabilities in walking (aOR: 1.31; 95% CI: 1.11, 1.54) and self-care (aOR: 1.58; 95% CI: 1.22, 2.05) had higher odds of anemia compared to their respective reference categories. Among the persons with diabetes, the respondents aged 80 years and above (aOR: 2.48; 95% CI: 1.56, 3.94), 70–79 years old (aOR: 1.38; 95% CI: 1.08, 1.76) and with disabilities in vision (aOR: 1.29; 95% CI: 1.02, 1.63) and walking (aOR: 1.50; 95% CI: 1.18, 1.91) were more likely to be anemic. Furthermore, among the older persons without diabetes, persons aged 80 years and above (aOR: 2.89; 95% CI: 2.05, 4.07), 70–79 years old (aOR: 1.46; 95% CI: 1.19, 1.80) and with difficulty in self-care (aOR: 1.87; 95% CI: 1.30, 2.69) were more likely to be anemic. Conclusions: The resilient predictors of developing anemia were advancing age and diabetes, based on our study. Anemia is significantly associated with walking and vision disabilities among older persons with diabetes and with self-care difficulties in those without diabetes. There is a need for future studies to evaluate strategies to prevent anemia among older adults in order to promote healthy aging.

## 1. Background

Anemia is common among older persons. Furthermore, it is increasingly prevalent, especially with advancing age [1]. Incidentally, the chronological age of 60 years is used as a cut-off mark for defining older persons in Malaysia [2]. Globally, the prevalence of anemia in those aged 60 years and above is 39%; it is 54.1% in Asia [3]. Anemia in the older persons is defined as a hemoglobin level of less than 13 gm/dL for men and 12 gm/dL for women, based on the WHO criteria [3]. The causes of anemia can be multifactorial in older persons. However, the most common causes of anemia in older persons are chronic disease and iron-deficiency anemia [4]. Anemia is also a risk factor for adverse outcomes such as hospitalization, morbidity and mortality [5,6,7,8,9,10].

Anemia in older persons is particularly relevant as it has a number of serious consequences. It is associated with higher incidences of cardiovascular disease [5] and cognitive impairment [6], decreased physical performance and quality of life [11,12,13], increased risk of falls, fractures, impaired muscle strength [11,14] and dementia [9], increased hospital admission and longer duration of hospitalization [15]. It has also been highlighted in the literature that anemia is associated with disabilities in activities of daily living (ADL) among hospitalized older persons [16]. The burden of disability is noted to increase in chronic diseases and aging populations [17]. Furthermore, the presence of anemia is significantly associated with longer hospital stays [5]. Chronic diseases such as diabetes mellitus and chronic kidney disease, which are prevalent among older persons, are also associated with anemia [18].

The National Health and Nutrition Examination Survey 2003–2012 mentioned that the anemia prevalence was highest in males aged 85 years old and over in the United States population [19]. The prevalence of anemia was also higher in the following groups of adults: (1) the oldest adults (>85 years old); (2) those with less education; (3) those with positive screening for cognitive decline; (4) those who reported a previous diagnosis of hypertension, diabetes, cancer, cardiovascular disease, encephalic vascular accident, osteoporosis, or three or more chronic diseases; and (5) those with depressive symptoms. However, these factors were not significantly related to their nutritional state in the study conducted by Corona LP et al. [18]. The authors [18] further mentioned that there are a few mechanisms that can explain the correlation between the prevalence of anemia with advancing age and chronic diseases. These include decreased renal hormone production, which leads to the development of anemia, and the increased expression of pro-inflammatory cytokines, which can contribute to erythropoietin insensitivity. Hence, the causes of anemia are multifactorial in nature among older persons. In terms of gender and ethnicity profiling, African Americans and Asians are more likely to develop anemia compared to Caucasians [20].

Studies on the association of anemia and chronic diseases and disabilities among older persons in Malaysia are limited. Moreover, chronic diseases are associated with complications. Furthermore, diabetes among older persons is associated with the burden of diabetes-related blindness due to diabetic retinopathy [21]. We know that anemia among older persons is caused by iron deficiency, chronic diseases and unexplained causes [22,23,24]. The three main factors that we examined in this study were socio-demography, chronic diseases and disability based on the conceptual framework in Figure 1.

## 2. Methods

### 2.1. Study Aim

The goal of this study was to determine the prevalence of anemia and its associated factors in terms of chronic diseases and disabilities among community-dwelling older persons in Malaysia.

### 2.2. Study Design

This cross-sectional study used the data from the National Health and Morbidity Survey 2015 (NHMS 2015) conducted by the Institute for Public Health, National Institute of Health and was funded by Ministry of Health, Malaysia. Detailed methodology of the survey is described in the technical report of NHMS 2015 [25]. The NHMS 2015 was a household survey involving Malaysian residents in non-institutional living quarters in both urban and rural areas from all 13 states and three federal territories (FTs) in Malaysia. The geographical locations were stratified into urban and rural to ensure representativeness during the sampling procedure. Two-stage stratified random-cluster sampling was applied. The primary sampling unit (PSU) was the enumeration block (EB) and the secondary sampling nit (SSU) was living quarters (LQs) within the selected EBs. The EB selection was conducted by the Department of Statistics Malaysia due to national data privacy.

### 2.3. Data Collection

The data were collected via face-to-face interviews using a mobile device. The NHMS questionnaires were used to gather data on demographic characteristics such as age, ethnicity, gender, marital status, education level and location by trained data collectors, including the nurses who conducted the clinical procedures [25]. Each capillary blood sample was tested for hemoglobin level using a HemoCue hemoglobinometer (HemoCue^®^ Hb 201+ System, Angelhom, Sweden) [26]. The capillary blood samples were also tested for fasting blood glucose and cholesterol level via CardioChek^®^ PA analyzer [27]. Blood pressure measurements were performed using Omron Japan Model HEM-907 equipment. Anthropometric measurements were obtained using Tanita Personal Scale HD 319 and SECA Stadiometer 213. The Washington Group on Disability Statistics (WG) was administered to identify people in the population with an increased risk of experiencing limited participation in society based on six domains [28]. All methods were carried out in accordance with relevant guidelines and regulations of conducting research and handling biospecimens and clinical waste. Informed consent was obtained from all participants prior to the survey.

### 2.4. Variables and Measures

Anemia among older persons was the main outcome of this study. The fact that the NHMS 2015 survey included non-communicable diseases (NCDs) such as hypertension, diabetes and hypercholesterolemia, as well as disabilities, was the guiding reason for selecting the explanatory variables in this study to determine the associations between anemia and NCDs with disabilities

The socio-demographic variables consisted of age groups in three categories: 60–69 years, 70–79 years and 80 years old and above. Ethnicity was classified into Malays and non-Malays. The non-Malays included Chinese and Indian respondents, along with other Bumiputera, comprising local Sabahans and Sarawakians, as well as others, who were mostly Malaysian citizens from indigenous populations. Education levels were classified as primary and lower education for those who had completed 6 years of primary school, or secondary and higher education for any education after completion of year 6 of primary school. Other socio-demographic variables were gender, marital status and location. Clinical variables included hypertension, diabetes, hypercholesterolemia and difficulties in sight, hearing, walking, self-care, remembering and communicating. Anemia was defined as a blood hemoglobin level below 12 g/dL for older women and below 13 g/dL for older men [29]. Respondents were considered as having hypertension if they were diagnosed with hypertension by medical personnel prior to the survey or if they had a resting blood pressure of 140/90 mm Hg or more. The presence of diabetes was determined based on a fasting blood glucose measurement of 6.1 mmol/L and above or random blood glucose measurement of 11.1 mmol/L and above. Respondents who were previously diagnosed with diabetes mellitus by a physician or assistant medical officer (AMO) reported themselves as “known diabetes”. For this survey, “known hypercholesterolemia” was defined as self-reported or diagnosis with hypercholesterolemia by a doctor or AMO. Respondents who had a total blood cholesterol of 5.2 mmol/L or more had hypercholesterolemia.

Disability was defined as having difficulty in at least one of the six following domains: sight, hearing, walking, cognition, self-care and communication. It was scored as having “a lot of difficulty or unable to do at all” or at least “some difficulty” in at least two domains. The cut-off was based on the definition used in Zambia [30] and in a disability paper among adults in Malaysia based on NHMS 2015 [31]. A review on the various models of disability and measurement methods concluded that the WG is a valid measure and is consistent with the International Classification of Functioning, Disability and Health (ICF) conceptual framework [28].

### 2.5. Data Analysis

The data on anemia among older persons in Malaysia were further analyzed using the IBM Statistical Package of Social Sciences (SPSS) for Windows version 23.0 (IBM Corp., Armonk, NY, USA). Descriptive analysis was performed to determine the socio-demographic distribution of the respondents. The estimated prevalence of anemia among older persons based on demography, chronic diseases and disability factors was determined using complex sample estimation, which applied the weight–age for each selected respondent. Categorical data were compared using Rao–Scott adjusted chi-square statistic for test of independence. Finally, all variables with a *p*-value of 0.25 or lower in the simple logistic regression model were included in a multivariable binary logistic regression model. The first model was adjusted for age for all the socio-demographic variables, chronic diseases and disabilities. The final model was further split into respondents with diabetes and respondents without diabetes because of the presence of significant interaction between diabetes and walking disabilities. The findings were presented as adjusted odds ratios (aORs) with 95% confidence intervals (CI), and a *p*-value of less than 0.05 was considered significant. Model fitness was assessed with Hosmer–Lemeshow goodness-of-fit test, with *p*-value higher than 0.05 indicating the model was fitted well.

## 3. Results

### 3.1. Descriptive Statistics

The overall response rate for the NHMS, 2015 survey was 86.4% (29,460 respondents); however, the response rate for the anemia module among older persons was 93.7% from the total of 3794 respondents. Thus, 3556 older persons were included in this study. Table 1 summarizes the sociodemographic characteristics of the study population. This study showed that 62.6% of the respondents were in the 60–69 years’ age group. There were more female respondents compared to males (53.5% vs. 46.5%), while Malays outnumbered non-Malays (64.8% vs. 35.2%). In addition, more than half of the respondents were married (67.7%), lived in rural areas (52.2%) and had primary and lower education (73.2%). The majority of the respondents had hypertension (70.8%) and hypercholesterolemia (68.1%). Furthermore, the majority of the older persons had no difficulty in seeing (62.2%), hearing (77.9%), walking (60.8%), remembering (75.6%), managing their self-care (90.2%) or communicating (89.0%).

### 3.2. Factors Associated with Anemia

The mean age of the respondents with anemia was 70 years, with 95% CI of 69.3–70.6 years. The prevalence of anemia among the older persons was 35.3% (95% CI: 33.1, 37.4). Among the older persons with diabetes, hypertension and hypercholesterolemia, the anemia prevalence rates were 38.6%, 35.3% and 34.1%, respectively. The anemia prevalence increased with advancing age; more than half (52.9%; 95% CI: 45.3, 60.4) of the respondents above 80 years of age were anemic. The bivariate analysis showed that most of the variables were significantly associated with anemia (*p* < 0.05), except for ethnicity, location, hypertension and hypercholesterolemia. The detailed prevalence rates and associations are shown in Table 2.

### 3.3. Multivariable Analysis of Factors Associated with Anemia

After adjusting for age (Appendix A), the association remained significant for increasing age, diabetes and disabilities related to walking, as well as self-care, in the multiple logistic regression analysis. The older persons aged 80 years and above were found to be 2.64 times more likely to be anemic (aOR: 2.64; 95% CI: 2.00, 3.47), while those aged 70–79 years old were 1.42 times more likely to be anemic (aOR: 1.42; 95% CI: 1.21, 1.66) compared to those aged between 60 and 69 years old. Those with diabetes were 1.3 times more likely to be anemic (aOR: 1.30; 95% CI: 1.13, 1.51) compared to those without diabetes. Respondents with walking disabilities were 1.31 times more likely to be anemic (aOR: 1.31; 95% CI: 1.11, 1.54) compared to those without a walking disability. The self-care-disability respondents were 1.58 times more likely to be anemic (aOR: 1.58; 95% CI: 1.22, 2.05) compared to those without self-care disabilities.

Furthermore, the model (Appendix A) was split into older persons with diabetes and those without diabetes.

Among those with diabetes, those in the age group of 80 years and above were 2.48 times more likely to be anemic (aOR: 2.48; 95% CI: 1.56, 3.94), while those aged 70–79 years old were 1.38 times more likely to be anemic (aOR: 1.38; 95% CI: 1.08, 1.76) compared to those aged between 60 and 69 years old. In addition, the older persons with disabilities in sight were 1.29 times more likely to be anemic (aOR: 1.29; 95% CI: 1.02, 1.63) compared to those without, whereas those with disabilities in walking were 1.50 times more likely to be anemic (aOR: 1.50; 95% CI: 1.18, 1.91) compared to those without.

Among the respondents without diabetes, those in the age groups of 80 years and above and 70–79 were found to be 2.89 times (aOR: 2.89; 95% CI: 2.05, 4.07) and 1.46 times (aOR: 1.46; 95% CI: 1.19, 1.80) more likely to be anemic, respectively, than those aged 60–69 years old. Furthermore, older persons with disabilities in self-care were 1.87 times more likely to be anemic (aOR: 1.87; 95% CI: 1.30, 2.69) compared to those without self-care difficulties. Appendix A insert here.

## 4. Discussion

This study demonstrated that the prevalence of anemia among older persons aged 60 years and above was 35.3%. This finding was attributed to the higher prevalence of co-morbidities, such as hypertension, diabetes and chronic kidney disease, in the older population [25]. Additionally, anemia in older persons in Malaysia was associated with Indian and Malay ethnicities, increasing age, hospitalization and diabetes in a prior study by Yusof et al. [32].

Chronic disease profiling showed that the prevalence rates of anemia in the older persons with diabetes, hypertension and hypercholesterolemia were 38.6%, 35.3% and 34.1%, respectively. However, only diabetes remained significant in the multivariate analysis. The older persons with diabetes were more likely to have anemia due to the increased risk of renal disease [18,33,34], especially chronic kidney disease. It is likely that interactions between the environment/lifestyle and genetic factors provide the explanation for the high risk of developing type 2 diabetes in this population. However, demonstrating such interactions is highly challenging. In addition, the increasing number of older persons and low death rates increase the proportion of people living with diabetes, rendering a large number of people at risk of acquiring this sequela [35]. Moreover, 20% of diabetics may develop nephropathy, according to the American Diabetes Association [36].

Furthermore, our study revealed that anemia prevalence increases with increasing age in older persons. This finding was in accordance with the literature, where it was found that anemia prevalence exceeds 20% in those aged 85 years and above [1]. The respondents aged 80 years and above were more likely to be anemic, together with the respondents aged between 70 and 79 years, compared to their younger counterparts in our study. These result are probably related to co-morbidities such as diabetes mellitus and chronic kidney disease in the older population, as was mentioned in other studies [25,37]. Furthermore, in the multiethnic Asian population, anemia was adversely associated with frailty, decreased muscle strength and the impairment of instrumental activities of daily living (IADL) [38]. However, we did not assess frailty, muscle strength or IADL impairment in our study.

We found that anemia in older persons was significantly associated with disabilities in activities such as self-care and walking, on multivariate analysis. This finding was similar to an observation in a previous study on older persons with anemia in Korea, who were found to be more likely to have abnormal Timed Up and Go (TUG) test results, which is related to walking disability, than individuals without anemia [39]. In addition, the disabilities that were significantly associated with older persons having diabetes were walking and sight. We hypothesized that the walking disabilities may have been due to the complications of diabetes, such as diabetic foot ulcers, while the sight difficulties could have been due to retinopathy. Furthermore, diabetic patients with retinopathy have lower levels of hemoglobin and a higher frequency of anemia [40]. The prevalence of diabetic retinopathy was reported to be 31% in Denmark, and at least 4% of these individuals seemed to have visual disturbances [41].

It was observed that, among the older persons without diabetes, self-care disability was significantly associated with anemia. This result was possibly due to other chronic diseases, such as dementia, among these respondents. In addition, there is accumulating evidence from population-based studies that anemia is independently associated with disability, as well as decreased physical performance and muscle strength [13], a predictor of cognitive and physical decline [6], as well as mortality [1,6].

Our findings consistently demonstrated that different disabilities were associated with older persons with diabetes and without diabetes.

## 5. Strengths and Limitations

This study estimated the prevalence of anemia and its associated factors in a nationally representative sample of older persons in Malaysia. Additionally, this is the first paper highlighting anemia and its association with chronic diseases and disability among older Malaysians. The sample size is adequate to produce a statistically sound finding. The data for this study were extracted from the NHMS 2015. The limitations of our study were mainly methodological due to the study’s cross-sectional design. Since it was a community survey, other existing gaps identified were the unavailability of staple food intake history, eating habits and nutritional intake. The data are exclusively Malaysian; thus, they may not offer an accurate representation, which limits the study’s generalization.

## 6. Conclusions

The resilient predictors of developing anemia were advancing age and diabetes, based on our study. Anemia is significantly associated with walking and vision disabilities among older persons with diabetes and with self-care difficulties in those without diabetes. There is a need for future studies to evaluate strategies to prevent anemia among older adults in order to promote healthy aging.

## Figures and Tables

**Figure 1 ijerph-19-04983-f001:**
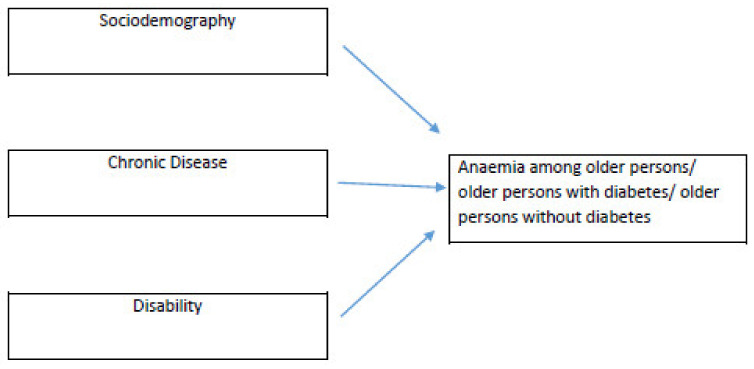
Conceptual framework predicting anemia based on socio-demography, chronic disease and disability among older persons.

**Table 1 ijerph-19-04983-t001:** Background characteristics of the study participants (N = 3556).

Characteristics of Respondents	Total (*n*)	%
**Age**		
60–69 years	2227	62.6
70–79 years	1052	29.6
≥80 years	277	7.8
**Gender**		
Male	1652	46.5
Female	1904	53.5
**Ethnicity**		
Malay	2305	64.8
Non-Malay	1251	35.2
**Marital Status**		
Single/widow/widower/divorcee	1149	32.3
Married	2407	67.7
**Location**		
Urban	1700	47.8
Rural	1856	52.2
**Education Level**		
Primary and lower	2569	73.2
Higher education	955	26.8
**Hypertension**		
Yes	2517	70.8
No	1039	29.2
**Diabetes**		
Yes	1404	39.5
No	2152	60.5
**Hypercholesterolemia**		
Yes	2422	68.1
No	1134	31.9
**Difficulty in Seeing**		
Yes	1343	37.8
No	2212	62.2
**Difficulty in Hearing**		
Yes	784	22.1
No	2770	77.9
**Difficulty in Walking**		
Yes	1395	39.2
No	2160	60.8
**Difficulty in Remembering**		
Yes	867	24.4
No	2687	75.6
**Difficulty in Self-Care**		
Yes	350	9.8
No	3204	90.2
**Difficulty in Communicating**		
Yes	399	11.0
No	3156	89.0

**Table 2 ijerph-19-04983-t002:** Prevalence of anemia among older persons according to chronic disease and disability factors: NHMS 2015.

Parameters	Anemia (*n* = 1311)				No Anemia (*n* = 2245)				*p* Value
	*n*	%	95% CI		*n*	%	95% CI		
			LL	UL			LL	UL	
Overall Anemia	1311	35.3	33.1	37.4	2245	64.7	62.6	66.9	
Mean age; years	-	70.0	69.3	70.6	-	67.9	67.4	68.3	
**Age**									
60–69 years	710	30.8	28.4	33.3	1517	69.2	66.7	71.6	
70–79 years	437	40.0	35.6	44.5	615	60.0	55.5	64.4	
≥80 years	164	52.9	45.3	60.4	113	47.1	39.6	54.7	**<0.001**
**Gender**									
Male	581	32.9	30.1	35.8	1071	67.1	64.2	69.9	
Female	730	37.6	34.5	40.7	1174	62.4	59.3	65.5	**0.025**
**Ethnicity**									
Malay	852	36.9	34.0	39.9	1453	63.1	60.1	66.0	
Non-Malay	459	33.7	30.6	36.9	792	66.3	63.1	69.4	0.142
**Marital status**									
Currently married	832	33.4	30.9	35.9	1575	66.6	64.1	69.1	
Currently not married	479	39.4	35.5	43.4	670	60.6	56.6	64.5	**0.010**
**Location**									
Urban	620	35.3	32.6	38.1	1080	64.7	61.9	67.4	
Rural	691	35.3	32.3	38.4	1165	64.7	61.6	67.7	0.996
**Education Level**									
Primary and below	990	37.1	34.6	39.7	1579	62.9	60.3	65.4	
Secondary and above	306	31.1	27.4	35.2	649	68.9	64.8	72.6	**0.012**
**Hypertension**									
Yes	940	35.3	32.7	37.9	1577	64.7	62.1	67.3	
No	371	35.3	31.5	39.2	668	64.7	60.8	68.5	0.992
**Diabetes**									
Yes	569	38.6	35.2	42.2	835	61.4	57.8	64.8	
No	742	33.1	30.5	35.8	1410	66.9	64.2	69.5	**0.011**
**Hypercholesterolemia**									
Yes	865	34.1	31.6	36.7	1557	65.9	63.3	68.4	
No	446	37.5	33.7	41.5	688	62.5	58.5	66.3	0.152
**Difficulty in Seeing**									
Yes	543	38.7	35.5	42.0	800	61.3	58.0	64.5	
No	767	33.0	30.2	36.0	1445	67.0	64.0	69.8	**0.012**
**Difficulty in Hearing**									
Yes	326	39.5	35.1	44.2	458	60.5	55.8	64.9	
No	984	34.0	31.5	36.6	1786	66.0	63.4	68.5	**0.045**
**Difficulty in Walking**									
Yes	613	41.9	38.5	45.4	782	58.1	54.6	61.5	
No	697	30.6	27.9	33.3	1463	69.4	66.7	72.1	**<0.001**
**Difficulty in Remembering**									
Yes	366	41.1	37.0	45.3	501	58.9	54.7	63.0	
No	943	33.1	30.5	35.7	1744	66.9	64.3	69.5	**0.001**
**Difficulty in Self-Care**									
Yes	193	54.1	47.2	60.9	157	45.9	39.1	52.8	
No	1117	33.2	31.0	35.5	2087	66.8	64.5	69.0	**<0.001**
**Difficulty in Communicating**									
Yes	189	42.7	36.4	49.2	210	57.3	50.8	63.6	
No	1121	34.2	31.9	36.6	2035	65.8	63.4	68.1	**0.013**

UL = upper limit, LL = lower limit for 95% confidence interval for the weighted prevalence (%). *p* values < 0.05 is significant and are in bold fonts.

## Data Availability

The data that support the findings of this study are available from the Director General of Health Malaysia, but restrictions apply to the availability of these data, which were used under license for the current study and are therefore not publicly available. Data are available, however, from the corresponding author upon reasonable request and with permission from the Director of Health Malaysia.

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
