# Peer review of "The Prevalence of Anemia and Its Associated Factors among Older Persons: Findings from the National Health and Morbidity Survey (NHMS) 2015"

_ijerph, 2022, doi:10.3390/ijerph19094983_

Round 1

Reviewer 1 Report

This manuscript describes the prevalence of anemia with unknown origin among the older people in Malaysia, with some influence by diabetes, and physical and psychoneurological disability (frailty).

Important findings of this manuscript are;

1:  Ethnic or racial difference among the Malays (including some Chinese and Indians) and the Americans (Caucasians, African Americans, Latinos, Asians) should be discussed, as the racial difference within United States are reported in which the Asians have the lowest prevalence of anemia.

2:  Contribution of comorbid physical disorders to idiopathic anemia, especially diabetes mellitus.

3:  Contribution of comorbid geriatric conditions, decline of both physical and neuropsychological conditions.

However, such findings are not well described, or not well discussed.

The first issue was not described in this manuscript. Refer to the published papers (Patel KV, et al. Semin Hematol. 2008 Oct; 45(4): 210–217. Stauder R, et al. Blood. 2018 Feb 1;131(5):505-514. Varghese JS, et al. Indian J Med Res. 2019 Oct; 150(4): 385–389.)

The second issue might be complicated. As chronic kidney disease is a well-known condition causing anemia. Prevalence of diabetes mellitus is extremely high in this study (39.5%). 20% of diabetics may develop nephropathy (American Diabetes Association. Diabetes Care 2004;27(suppl 1):s79–s83). As the chronic kidney disease is an important factor for anemia, contribution of the high prevalence of diabetes in this population to the anemia should be discussed. Also, the prevalence of diabetic retinopathy is reported to be 31% in Denmark (Hove MN, et al. Acta Ophthalmol Scand. 2004 Aug;82(4):443-8), and at least 4% of them seem to have visual disturbance. Such subjects, diabetics with visual disturbances, might nephropathy, which may lead to anemia. Such aspect should be discussed.

Impairment in the daily life issue is categorized as “frailty.” Recent publication from Singapore (Lee CT, et al. J Nutr Health Aging. 2021;25(5):679-687.) discussed such aspect. The authors should reconsider such aspect.

In general, the findings in this study is important, however, more detailed discussions regarding the epidemiology and the pathogenetic factors are required.

List of the references should follow the publisher’s instructions (https://www.mdpi.com/authors/layout#_bookmark93).

Author Response

1) Response to reviewer 1,2,3

2)ijerph review with track changes

3)ijerph review edited 

Response to reviewer 1

  1. Ethnic or racial difference among the Malays (including some Chinese and Indians) and the Americans (Caucasians, African Americans, Latinos, Asians) should be discussed, as the racial difference within United States are reported in which the Asians have the lowest prevalence of anemia.

Regarding comment 1

Thank you for the relevant comments however, I wish to clarify that this paper focuses on the association of anaemia and chronic diseases as well as disabilities among the older persons in Malaysia. It does not focus on the ethnic differences of the elderly population and anaemia as this area has been extensively discussed in an earlier publication utilizing the similar data set by Yusof M et al[32]

I have added Line 238-240 to highlight the prior study covering ethnicity and anaemia among older Malaysians.

Additionally, anaemia in the older persons in Malaysia was associated with Indian and Malay ethnicities, increasing age, hospitalization, and diabetes in a prior study by Yusof M et al [32].

  1. Contribution of comorbid physical disorders to idiopathic anemia, especially diabetes mellitus.

In general, the findings in this study is important, however, more detailed discussions regarding the epidemiology and the pathogenetic factors are required.

Regarding comment 2

Thank you for the comments, I have added Line 244-251 to support this comment

Older persons with diabetes were more likely to have anemia due to the increased risk of renal disease [33, 34, 18] especially chronic kidney disease. It is likely that interactions between the environment/lifestyle and genetic factors provides the explanation for the high risk of developing type 2 diabetes in this population. However, demonstrating such interactions is highly challenging. In addition, the increasing ageing persons and low death rates will increase the proportion of people living with diabetes, rendering a large number of people at risk of acquiring this sequelae [35]. Moreover, 20% of diabetics may develop nephropathy according to the American Diabetes Association [36].

  1. Contribution of comorbid geriatric conditions, decline of both physical and neuropsychological conditions.

Regarding comment 3

Thank you for the comments

I have addressed this as below in Line 258-261

Besides, in the multiethnic Asian population, anemia was adversely associated with frailty, decreased muscle strength, and instrumental activities of daily living(IADL) im-pairment [40] . However, we did not assess frailty, muscle strength and IADL impairment in our study.

  1. Detailed discussion regarding the epidemiology and pathogenetic factors are required

I have address this comment in Line 238-240

Additionally, anaemia in the older persons in Malaysia was associated with Indian and Malay ethnicities, increasing age, hospitalization, and also diabetes in a prior study by Yusof M et al [32].

  1. List of the references should follow the publisher’s instructions

(https://www.mdpi.com/authors/layout#_bookmark93).

               This has been updated under references

Reviewer 2 Report

My suggestion is that the discussion should be further explored by comparing the findings with similar studies in different populations and proposing a new interpretation or study.

Author Response

Please see the attachment on

1) response to reviewer 1,2 & 3

Response to Reviewer 2

My suggestion is that the discussion should be further explored by comparing the findings with similar studies in different populations and proposing a new interpretation or study.

Thank you for the comments. I have added several studies for this comment in

Line 263-266 and

This finding was similar to that of older persons in Korea with anaemia, whereby they are more likely to have abnormal Timed Up and Go Test (TUG test) results which is related to walking disability than individuals without anaemia [41].

Line269-272 and

Furthermore, diabetic patients with retinopathy have lower level of hemoglobin and high-er frequency of anemia [37]. The prevalence of diabetic retinopathy was reported to be 31% in Denmark and at least 4% of them seem to have visual disturbances [38].

Line 275-278

In addition, there are accumulating evidence from population-based studies that anaemia is independently associated with disability, decreased physical performance together with muscle strength [13], a predictor for cognitive with physical decline [6], and mortality [6,1].

Reviewer 3 Report

It is an interesting paper in its results and conclusions, although I see two problems:
- The data collected are from 2015, which gives us data that could be obsolete, especially considering that we have gone through a pandemic and it could be that the data have undergone variation.
- Malaysia is 74th in the life expectancy ranking, the question is whether the data obtained can be exported to other parts of the world with a higher life expectancy.

With more current data, it would be an interesting paper.

Author Response

I have enclosed the response to the reviewer as below:

1) response to reviewer 1,2,3

Response to reviewer 3

The data collected are from 2015, which gives us data that could be obsolete, especially considering that we have gone through a pandemic and it could be that the data have undergone variation.

With more current data, it would be an interesting paper.

Thank you for the comment, however we disagree that the data could be obsolete since the National Health and Morbidity Survey (NHMS) is a constantly on-going survey with regards to data collection. We will be happy to publish the 2018 data on this topic once it is available.

Line 283-285

In addition, the subject of this research is still clinically relevant and valid as it is the first paper from Malaysia highlighting anaemia and its association with chronic diseases together with disability.

Malaysia is 74th in the life expectancy ranking, the question is whether the data obtained can be exported to other parts of the world with a higher life expectancy.

Thank you for this comment, I agree this is a limitation of our study and I have included this under the limitation of this study in Line 287-289

This is an exclusively Malaysian data, thus it may not be an accurate representation and this limits its generalization.

Round 2

Reviewer 3 Report

I understand the authors' comments and accept them.